# Spectroscopic visualization and phase manipulation of chiral charge density waves in 1T-TaS$_2$

Yan Zhao[1,2,8], Zhengwei Nie[3,4,8], Hao Hong[5], Xia Qiu[2,4,6], Shiyi Han[1], Yue Yu[1], Mengxi Liu [4,6], Xiaohui Qiu[4,6], Kaihui Liu [5], Sheng Meng [3,4,7] ✉, Lianming Tong [1] ✉ & Jin Zhang [1]

The chiral charge density wave is a many-body collective phenomenon in condensed matter that may play a role in unconventional superconductivity and topological physics. Two-dimensional chiral charge density waves provide the building blocks for the fabrication of various stacking structures and chiral homostructures, in which physical properties such as chiral currents and the anomalous Hall effect may emerge. Here, we demonstrate the phase manipulation of two-dimensional chiral charge density waves and the design of in-plane chiral homostructures in 1T-TaS$_2$. We use chiral Raman spectroscopy to directly monitor the chirality switching of the charge density wave—revealing a temperature-mediated reversible chirality switching. We find that interlayer stacking favours homochirality configurations, which is confirmed by first-principles calculations. By exploiting the interlayer chirality-locking effect, we realise in-plane chiral homostructures in 1T-TaS$_2$. Our results provide a versatile way to manipulate chiral collective phases by interlayer coupling in layered van der Waals semiconductors.

Charge density wave (CDW) refers to the periodic modulation of electronic distributions accompanied by the lattice distortion[1–4]. In recent years, the chirality of CDW has attracted considerable interest due to its importance for superconductivity[5–7], magnetism[8–10], and potential applications in memory devices and sensors[11–13]. In previous reports, chiral CDW has been revealed in 1T-TiSe$_2$[14–16], Ti-doped 1T-TaS$_2$[17], and kagome superconductor KV$_3$Sb$_5$[10], which has been construed to be a chiral electronic charge order in terms of a combined charge and orbital order[18], and the helicity of CDW is a three-dimensional (3D) one formed in the stacking direction. Recently, two-

dimensional (2D) chiral CDW has been observed in 1T-TaS$_2$[19] and 1T-NbSe$_2$[20], whose in-plane mirror symmetries are broken as the CDW phase transition occurs. The 2D chiral CDW provides a platform to fabricate diverse 3D stacking structures or design in-plane chiral homostructures (CHS) on demand, where unprecedented physical phenomena may appear. However, the 3D stacking configuration of chiral CDW has not been reported, and more importantly, although in-plane chiral phase transition can be triggered by laser pulse[19], electric field[20], and magnetic field[10,21], the in-plane CHS of CDW has not been realized yet.

[1]College of Chemistry and Molecular Engineering, Beijing Science and Engineering Center for Nanocarbons, Beijing National Laboratory for Molecular Sciences, Peking University, Beijing 100871, P. R. China. [2]Academy for Advanced Interdisciplinary Studies, Peking University, Beijing 100871, P. R. China. [3]Beijing National Laboratory for Condensed Matter Physics and Institute of Physics, Chinese Academy of Sciences, Beijing 100190, P. R. China. [4]University of Chinese Academy of Sciences, Beijing 100049, P. R. China. [5]State Key Lab for Mesoscopic Physics and Frontiers Science Center for Nano-optoelectronics, Collaborative Innovation Center of Quantum Matter, School of Physics, Peking University, Beijing 100871, P. R. China. [6]CAS Key Laboratory of Standardization and Measurement for Nanotechnology, CAS Center for Excellence in Nanoscience, National Center for Nanoscience and Technology, Beijing 100190, P. R. China. [7]Songshan Lake Materials Laboratory, Dongguan, Guangdong 523808, P. R. China. [8]These authors contributed equally: Yan Zhao, Zhengwei Nie. ✉ e-mail: smeng@iphy.ac.cn; tonglm@pku.edu.cn

In this work, we explored the in-plane chiral CDW and its 3D stacking configuration in 1T-TaS$_2$, and further realized the on-demand fabrication of in-plane CHS by virtue of the interlayer chirality-locking effect. We found apparent chiral Raman response in the commensurate (C) and nearly commensurate (NC) CDW phases of 1T-TaS$_2$, which provides a convenient and direct method to identify the chirality of 1T-TaS$_2$. Then the reversible chirality switching of CDW stimulated by annealing cycles was unveiled. More importantly, the chirality of different layers in a 1T-TaS$_2$ flake remains the same in multiple annealing cycles due to the strong interlayer binding energy for the layers with the same chirality, which is referred to as interlayer chirality-locking effect. Utilizing the energy preference for same-chirality stacking, we realized the fabrication of in-plane CHS by vertically stacking two 1T-TaS$_2$ flakes with opposite chirality. The overlapped zone will transform from opposite-chirality stacking to same-chirality stacking after annealing, whereas the chirality of other parts of the flakes will not be affected, enabling the formation of chiral domain walls in one flake.

## Results

### Chiral Raman signal of 1T-TaS$_2$

The 1T-TaS$_2$ flakes (Fig. 1a) were mechanically exfoliated from the bulk crystal (HQ Graphene), and the thickness was measured to be 15 nm by the atomic force microscope (AFM) (Fig. 1b). Raman spectra excited by left-handed (σ+) and right-handed (σ−) circularly polarized light at room temperature are shown in Fig. 1c. Prominent Raman intensity difference for the σ+ and σ− excitation is observed, as extracted in Fig. 1d, reflecting clearly Raman optical activity of 1T-TaS$_2$. Further experiments show that the chiral Raman response commonly exists in 1T-TaS$_2$ flakes with different thicknesses for 532 nm and 633 nm laser excitation (Supplementary Fig. 1).

We then applied the helicity-resolved Raman spectroscopy (HRRS)[22] to identify the symmetry of each Raman mode (Fig. 1e). According to the Raman selection rule determined by group theory and symmetry analysis[23], the Raman active modes of 1T-TaS$_2$ with a trigonal crystal lattice are $E_g$ and $A_{1g}$ modes[24–26] (see detail in Supplementary Information Section I). The $E_g$ mode can only be detected for the cross-polarized (σ+σ− or σ−σ+) configuration whereas the $A_{1g}$ mode can only be detected for the co-polarized (σ+σ+ or σ−σ−) configuration (Supplementary Information Section II). Thus we can assign the symmetries of Raman modes in Fig. 1e for the NCCDW phase. The $A_{1g}$ Raman modes are at 69 cm$^{-1}$ ($A_{1g}^1$), 306 cm$^{-1}$ ($A_{1g}^2$), and 381.5 cm$^{-1}$ ($A_{1g}^3$). The $E_g$ modes are at 60 cm$^{-1}$ ($E_g^1$), 66 cm$^{-1}$ ($E_g^2$), 69.5 cm$^{-1}$ ($E_g^3$), 74 cm$^{-1}$ ($E_g^4$), 243 cm$^{-1}$ ($E_g^5$), 261 cm$^{-1}$ ($E_g^6$), 280 cm$^{-1}$ ($E_g^7$), and 295.5 cm$^{-1}$ ($E_g^8$). The HRRS of CCDW phase and the assignment of Raman modes are shown in Supplementary Fig. 5. It is worth pointing out that we can identify two $A_{1g}$ modes and five $E_g$ modes at the low-frequency 40–100 cm$^{-1}$ range by using HRRS for the

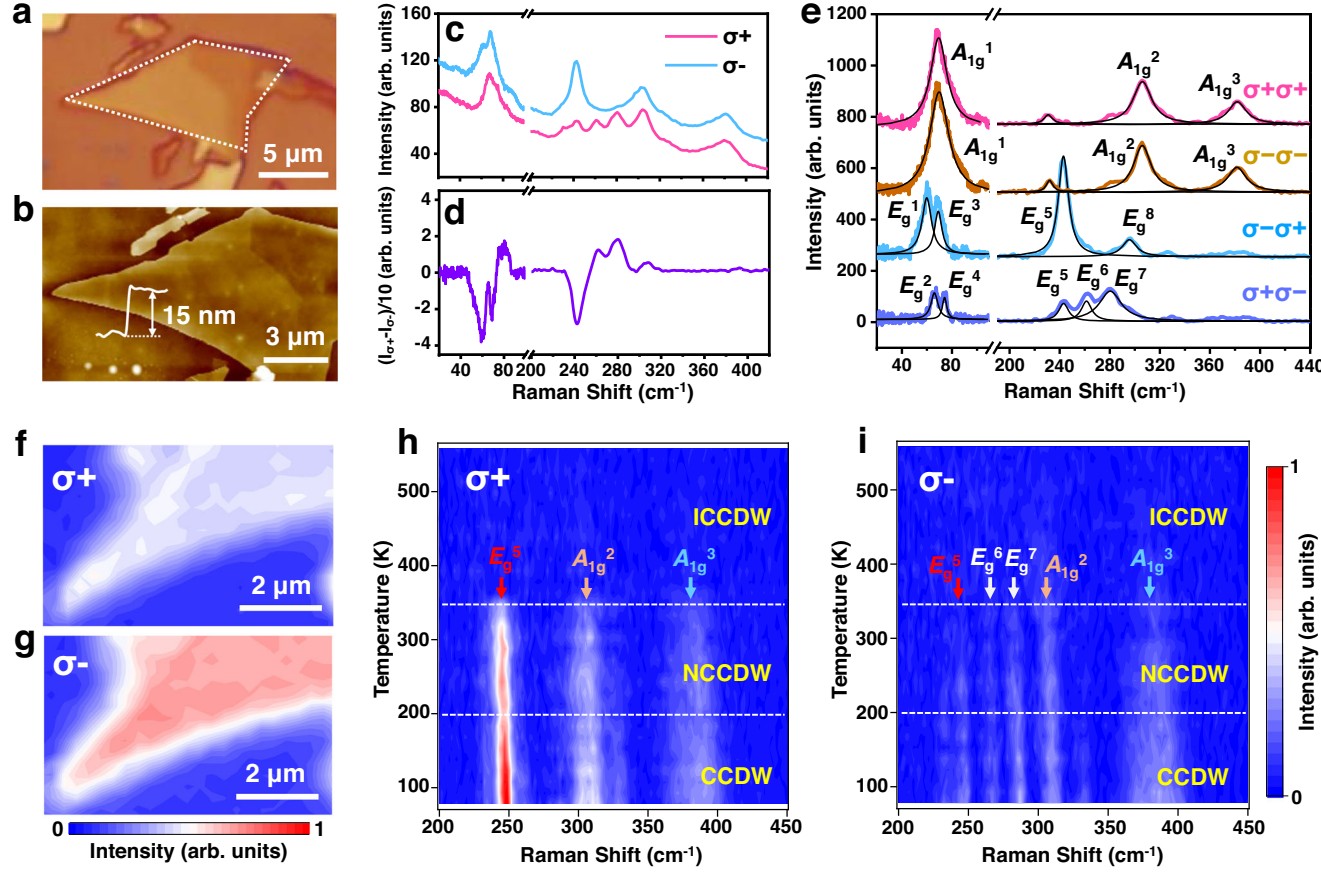

**Fig. 1 | Chiral Raman response of 1T-TaS$_2$ and its temperature dependence.**
**a** Optical picture of mechanically exfoliated 1T-TaS$_2$ flake. **b** AFM image and the thickness of the 1T-TaS$_2$ sample. **c** Chiral Raman spectra of 1T-TaS$_2$ excited by left-handed (σ+) and right-handed (σ−) circularly polarized light at room temperature. **d** Raman intensity difference (I$_{σ+}$−I$_{σ−}$) extracted from **c** after removing the background. **e** Helicity-resolved Raman spectra of 1T-TaS$_2$ at room temperature. **f**, **g** Raman intensity mapping of $E_g^5$ mode of the 1T-TaS$_2$ flake under the **f** σ+ and **g** σ− laser excitation. **h**, **i** False-color Raman spectra showing the variation of Raman intensity as a function of temperature under the **h** σ+ and **i** σ− excitation, which shows the chiral Raman response for different CDW phases. The white dotted lines approximately illustrate the temperature ranges for the commensurate (C), nearly commensurate (NC), and incommensurate (IC) CDW phases according to the variation of chiral Raman intensities. The red, yellow, and blue arrows indicate the $E_g^5$, $A_{1g}^2$, and $A_{1g}^3$ Raman modes, respectively. The two white arrows in **i** represent $E_g^6$ and $E_g^7$ modes, which can only be detected for σ− excitation. The wavelength of the excitation laser is 532 nm.

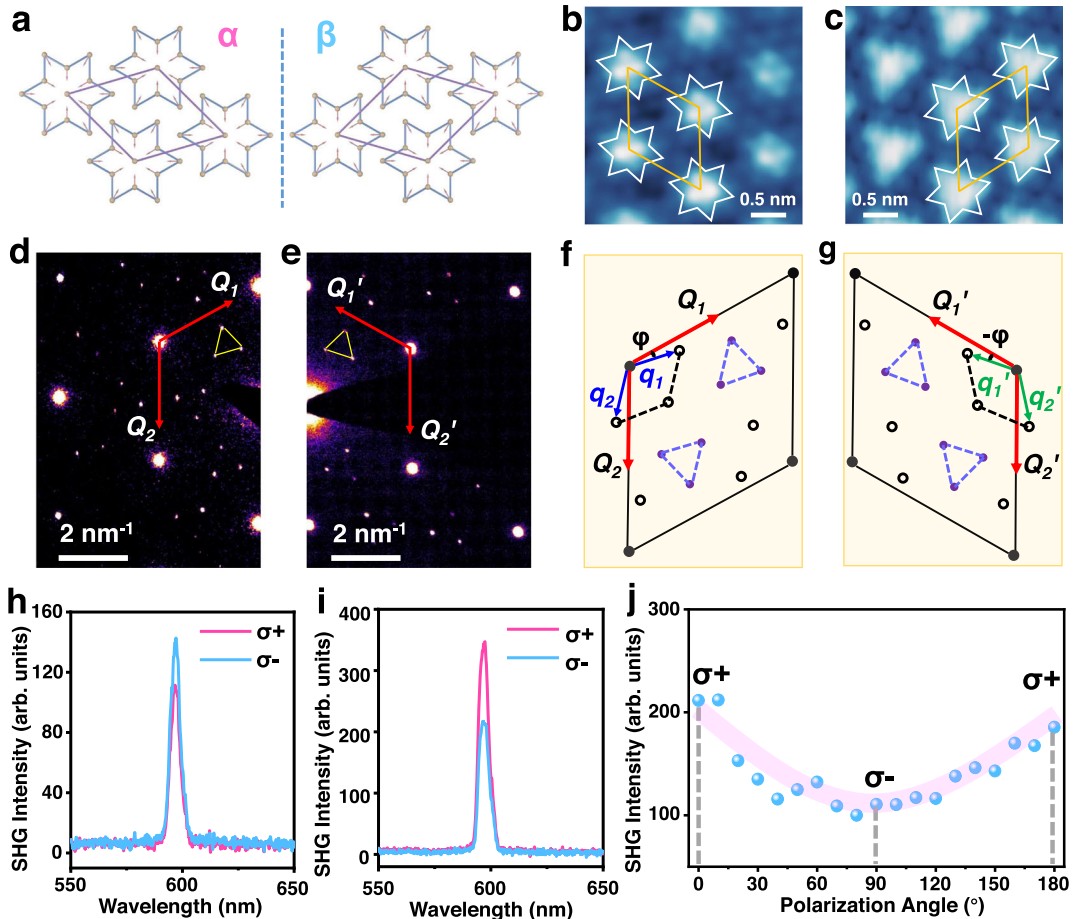

**Fig. 2 | Characterization of the chiral structure of 1T-TaS₂. a** Schematics of the two enantiomers (α and β) for the CDW superlattice structure. **b**, **c** The STM images of chiral CDW corresponding to α and β phases. The white star and yellow rhombus illustrate the David star structure and unit cell of the superlattice, respectively. **d**, **e** SAED results of two 1T-TaS₂ samples with opposite chiral Raman response. $Q_1$, $Q_2$ (or $Q_1'$, $Q_2'$) represent the reciprocal vectors of the atomic lattice. The diffraction spots in the triangular groups are from the CDW superlattice. **f**, **g** Schematic

illustration of the SAED patterns for the NCCDW phases with opposite chirality. Solid dots and circles represent the diffraction spots of atomic lattice and CDW superlattice structure, respectively. $q_1$, $q_2$ (or $q_1'$, $q_2'$) represent the reciprocal vectors of the CDW superlattice. φ and −φ are the rotation angles of the chiral CDW superlattices with respect to the atomic lattice. **h**, **i** SHG signals of two 1T-TaS₂ samples with opposite chirality. **j** Variation of SHG intensity as a function of the polarization angle of the incident light.

CCDW phase, rather than two $E_g$ modes as reported previously[27] (see detailed low-frequency Raman spectra in Supplementary Fig. 6 and Supplementary Fig. 7). Most importantly, we found the obvious dependence on the vibration mode symmetries for the chiral Raman response. The out-of-plane $A_{1g}$ Raman modes for the σ+σ+ and σ−σ− configurations are almost the same, whereas the intensities of in-plane $E_g$ modes differ remarkably for the σ+σ− and σ−σ+ configurations, indicating that the chiral Raman response may correspond to an in-plane chiral structure. Similar experimental results of chiral Raman response in 1T-TaS₂ have been reported recently, which were well explained by Raman tensor analysis, either by introducing an antisymmetric Raman tensor or including complex Raman tensor elements[28,29].

Figure 1f, g show the Raman intensity mapping of the $E_g^5$ mode of 1T-TaS₂ at room temperature under the σ+ and σ− excitation, demonstrating the uniform chiral Raman signal. 1T-TaS₂ in the undistorted phase and the incommensurate (IC) CDW phase show no chiral Raman signal (see detailed Raman spectra in Supplementary Fig. 8 and Supplementary Fig. 9) when the temperature is above ~350 K. Upon lowering the temperature, it transforms into NCCDW phase, and finally forms CCDW below ~200 K. Chiral Raman response prominently appears with the formation of NCCDW and CCDW phase, as shown in Fig. 1h, i. The phase transition temperature can be obtained from the variation of Raman intensities[27,30,31] (Supplementary Fig. 10), as labeled

by the white dotted lines. Besides, the hysteresis behavior of the chiral Raman response can also be identified (see detail in Supplementary Fig. 11).

## Characterization of the chiral CDW in 1T-TaS₂

For the CCDW phase of 1T-TaS₂, the in-plane superlattice exhibits a $\sqrt{13} \times \sqrt{13}$ periodic unit cell containing 39 atoms[1,5]. And for the NCCDW phase, there are also patches of commensurate domains separated by discommensurate networks[32–34]. However, the vertical mirror plane symmetries are broken for the commensurate superlattice, generating a planar chiral structure with a $C_3$ point group[35]. The two energetically equivalent configurations (named as α and β) are schematically shown in Fig. 2a. In Fig. 2b, c, we show the scanning tunneling microscopy (STM) images obtained at 77 K for the two enantiomers, which are measured in bulk 1T-TaS₂, revealing the chiral CDW states corresponding to the chiral superlattices. The selected area electron diffraction (SAED) patterns of two 1T-TaS₂ flakes transferred onto SiNₓ grid, which show opposite chiral Raman signals, were measured at room temperature and presented in Fig. 2d, e (see the detail of the prepared sample and scanning transmission electron microscopy (STEM) image in Supplementary Fig. 12). There is only one set of bright diffraction spots arranged hexagonally corresponding to the Bragg scattering from the crystal lattice[36–38], which reveals the single crystalline structure of the

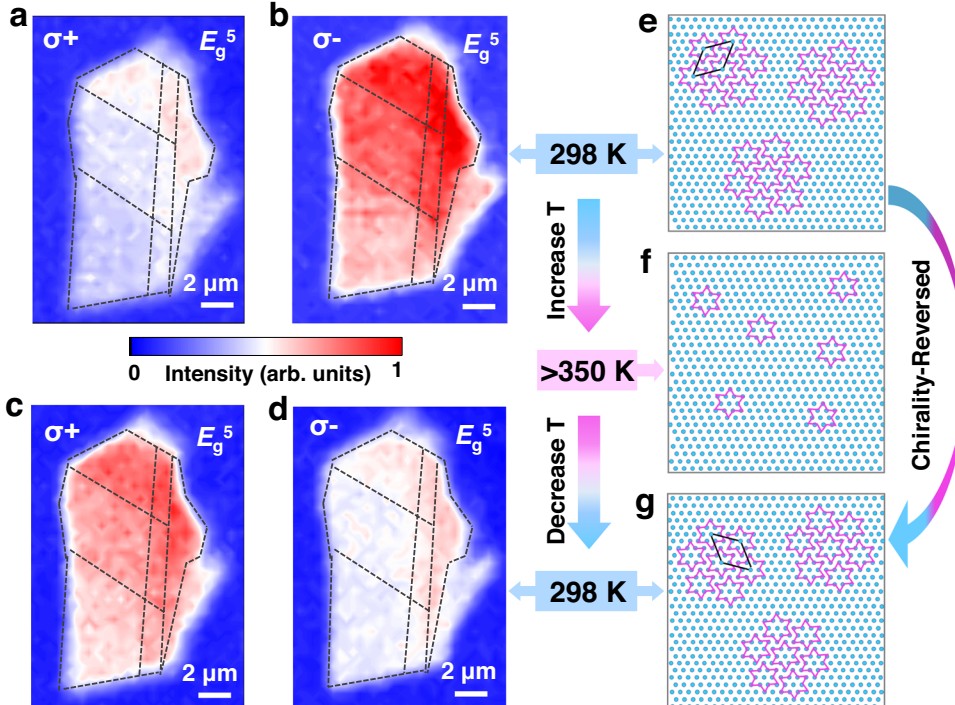

**Fig. 3 | Chirality switching of 1T-TaS$_2$ during the temperature variation process.** **a**, **b** Raman intensity mapping images for the $E_g^5$ mode of a multilayer 1T-TaS$_2$ flake under **a** σ+ and **b** σ− excitation at 298 K. The dotted lines divide the flake into several parts with different thicknesses. **c**, **d** Raman intensity mapping images for the $E_g^5$ mode of the chirality-reversed 1T-TaS$_2$ under **c** σ+ and **d** σ− excitation after the annealing cycle. The wavelength of the excitation laser is 532 nm. **e**–**g** Schematics of **e** initial NCCDW phase at the temperature (T) of 298 K; **f** ICCDW phase with T > 350 K (normal phase above ~550 K); **g** chirality-reversed NCCDW phase at 298 K after the annealing process. The unit cells of the chiral superlattice are illustrated by the black rhombus.

1T-TaS$_2$ domain. Besides, we can find the satellite diffraction spots appeared as triangular groups (yellow triangles) distributed around the bright diffraction spots, which correspond to the superlattice structure with periodic lattice distortion[1,39–41]. Most importantly, the orientation of the triangular groups relative to the $\mathbf{Q}_1$, $\mathbf{Q}_2$ (or $\mathbf{Q}_1'$, $\mathbf{Q}_2'$) reciprocal vectors of the atomic lattice are opposite for these two samples, consistent with the SAED results of the two enantiomers of 1T-TaS$_2$ shown in Fig. 2f, g, where $\mathbf{q}_1$, $\mathbf{q}_2$ (or $\mathbf{q}_1'$, $\mathbf{q}_2'$) are reciprocal vectors of the CDW superlattice. The new rhombic supercell can be rotated clockwise (with a rotation angle of φ) or anticlockwise (with a rotation angle of −φ) relative to the undistorted atomic unit cell[19], corresponding to the α or β chiral phase. It should be pointed out that we cannot distinguish the most adjacent first-order satellite diffraction spots around the apexes of the large rhombus (circles in Fig. 2f, g) in the experimental results of Fig. 2d, e, attributed to their weaker intensity for the NCCDW phase[34,41]. Nevertheless, we can conclude that the chiral Raman response originates from the chiral superlattice structure, and it can be used to conveniently differentiate the CDW phases with opposite chirality.

To further verify the mirror symmetry breaking and the induced chiral optical effect of 1T-TaS$_2$, we measured the second harmonic generation (SHG) spectrum of 1T-TaS$_2$ samples with opposite chirality and unveiled their SHG circular dichroism (SHG-CD) effect[42–44] (see detail in Supplementary Information Section III) under excitation of circularly polarized light (Fig. 2h, i). Figure 2j manifests the variation of SHG intensity as a function of the rotation angle of a quarter-wave plate in the incident light path, which clearly shows the SHG intensity difference for the σ+ and σ− circularly polarized excitation.

**Chirality switching and chiral stacking configuration of 1T-TaS$_2$**

We further explore the chirality switching during the temperature variation process. Figure 3a, b show the Raman intensity mapping

images of the $E_g^5$ mode of a terraced multilayer 1T-TaS$_2$ (see its optical picture and AFM image in Supplementary Fig. 13) for the σ+ and σ− excitation at 298 K, respectively. It can be seen that the Raman intensity under σ− excitation is uniformly larger than that under σ+ excitation, indicating the same chiral phase for the whole sample. Interestingly, after annealing above 350 K, where the chiral CDW domains are disrupted, the chiral Raman response is reversed as revealed by the Raman intensity mapping under σ+ and σ− excitation in Fig. 3c, d (see the chiral Raman spectra in Supplementary Fig. 14). The chirality switching process is illustrated in the schematic diagrams of Fig. 3e–g. It is important to notice that the chirality switching between α and β phases is random due to the energy degeneracy of the two chiral configurations (see results of multiple annealing cycles in Supplementary Fig. 15).

The remarkable chiral response of multilayer 1T-TaS$_2$ reveals that each layer may be stacked with the same chirality even during multiple chirality switching cycles, indicating the chirality-locking effect between layers. The SAED results of multilayer samples in Fig. 2d, e also show one set of diffraction spots, either clockwise or anticlockwise rotated, verifying the same chirality between layers. For 1T-TaS$_2$ layers stacked with the same chirality, it has been reported that a dimer stacking configuration is exceptionally stable[45–47]. Although the vertical stacking patterns of 1T-TaS$_2$ have long been discussed, the stacking order considering the degree of freedom of chirality has not been studied. We assume the chirality-locking effect between layers may originate from the interlayer interaction difference between same-chirality (α/α or β/β, Fig. 4a) and opposite-chirality (α/β, Fig. 4b) stacking configurations. Interlayer binding energy ($E_{ib}$), defined as the energy cost to separate van der Waals (vdW)-bonded atomic layers into isolated ones, i.e., $E_{ib} = (\sum E_{layer} - E_{total})/N_{cell}$, is used to characterize the strength of interlayer interactions. Using first-principles density functional theory (DFT) calculations, we calculated the $E_{ib}$ for both α/α (or β/β) and α/β stacking, demonstrating that $E_{ib}$ for α/α and

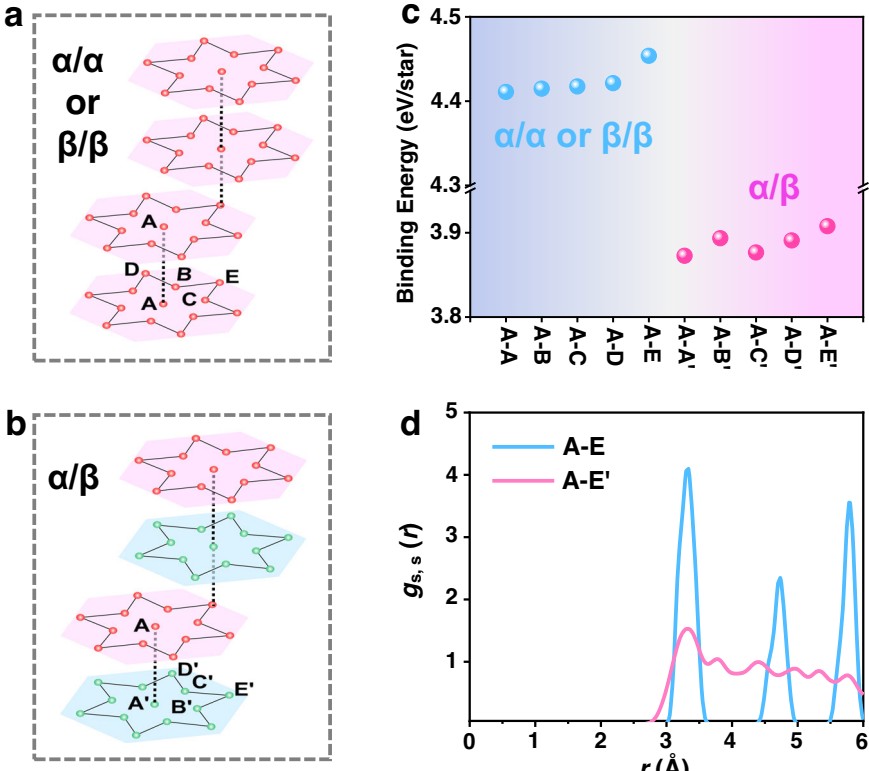

**Fig. 4 | Analysis of the interlayer interaction for the same-chirality and opposite-chirality stacking patterns. a, b** Schematics of the stacking orders of 1T-TaS$_2$ layers with **a** same chirality (take A-E stacking as an example) and **b** opposite chirality (take A-E' stacking as an example). The paired stacking pattern is adopted. **c** The binding energy ($E_{ib}$) for five stacking patterns of the same chirality (A-A, A-B, A-C, A-D, A-E) and five of the opposite chirality (A-A', A-B', A-C', A-D', A-E'). **d** The partial radial distribution function $g_{s, s}$ ($r$) for the same-chirality A-E stacking and opposite-chirality A-E' stacking. The two subscripts of $g_{s, s}$ ($r$) indicate sulfur atoms in adjacent layers, and $r$ indicates the averaged distance between them.

β/β (~4.4 eV/star) is much stronger than α/β (~3.9 eV/star, Fig. 4c), resulting in the stacking preference for the same-chirality order (see more detailed information about the stacking patterns and DFT results in Supplementary Information Section IV).

Moreover, we use the partial radial distribution function ($g(r)$), which describes how particle density varies as a function of distance from a reference particle, to reflect the distribution of sulfur atoms between layers and to further understand the origin of the $E_{ib}$ difference (see more detail of $g(r)$ in Supplementary Information Section IV). Figure 4d displays the calculated $g(r)$ for the same-chirality A-E stacking (the most stable same-chirality stacking configuration with a stacking vector $\mathbf{T}_S = -2\mathbf{a} + \mathbf{c}$, see detailed definition in Supplementary Information Section IV and Supplementary Fig. 16) and opposite-chirality A-E' stacking (see detailed definition in Supplementary Information Section IV and Supplementary Fig. 17). There are several sharp peaks for $g(r)$ of A-E stacking, indicating the regular distribution of the sulfur atoms between adjacent layers. In contrast, $g(r)$ of A-E' stacking shows a relatively constant curve oscillating around $g(r) = 1$, which indicates that the interlayer sulfur atoms distribute more randomly. More importantly, for the A-E' stacking, we can find the nonzero $g(r)$ below the average value of $r = 3$ Å, inducing strong instability of the system. The results above reveal the importance of interlayer chirality correlation in 1T-TaS$_2$ despite the weak bonding and reduced CDW coupling between layers as reported previously[48,49]. Finally, it is worth pointing out that we cannot directly compare the theoretical results with the experiment since the total energy of a given material configuration is not experimentally measurable. Besides, the present DFT calculations adopt an ideal model for the CCDW phase, whereas the presence of domains, strains, impurities,

defects, and imperfections are not included due to the unsurmountable computational costs. A further comprehensive study is needed to make the DFT results more precise and directly comparable with the experiments.

## Fabrication of in-plane chiral homostructure of CDW

Utilizing the chirality-locking effect between adjacent layers, we can manipulate the chiral CDW stacking and further design the in-plane CHS of CDW. Here we fabricated an out-of-plane chiral homostructure vertically stacked by two 1T-TaS$_2$ flakes with opposite chiral Raman response, and thus opposite chirality (named as α/β 1T-TaS$_2$), as shown in Fig. 5a. Figure 5b schematically depicts the energy diagram of α/β, α/α, and β/β stacking configurations. After annealing, the α/β stacking in the overlapped zone can transform into the more stable α/α or β/β stacking. As shown in Fig. 5c, d, the chiral Raman response of the overlapped zone vanishes for the primitive homostructure, whereas it appears after annealing at 473 K for 2 hours, which indicates that the opposite-chirality stacking has transformed to the same-chirality stacking for the overlapped zone.

Moreover, we note that a domain boundary between α and β phases is generated along the edges of the overlapped zone, which is apparently shown in the chiral Raman intensity mapping image of $E_g^5$ mode shown in Fig. 5e (also see that of the $E_g^6$ mode in Supplementary Fig. 18). This indicates that the relaxation of chiral phase in the overlapped zone will not affect the chirality of other parts of the two flakes. If the sample is annealed again above 350 K, the separated parts of the homostructure (i.e., the overlapped zone, the other parts of the lower or upper flake on the substrate) switch their chirality independently (Supplementary Fig. 19), which may be attributed to the strain field and

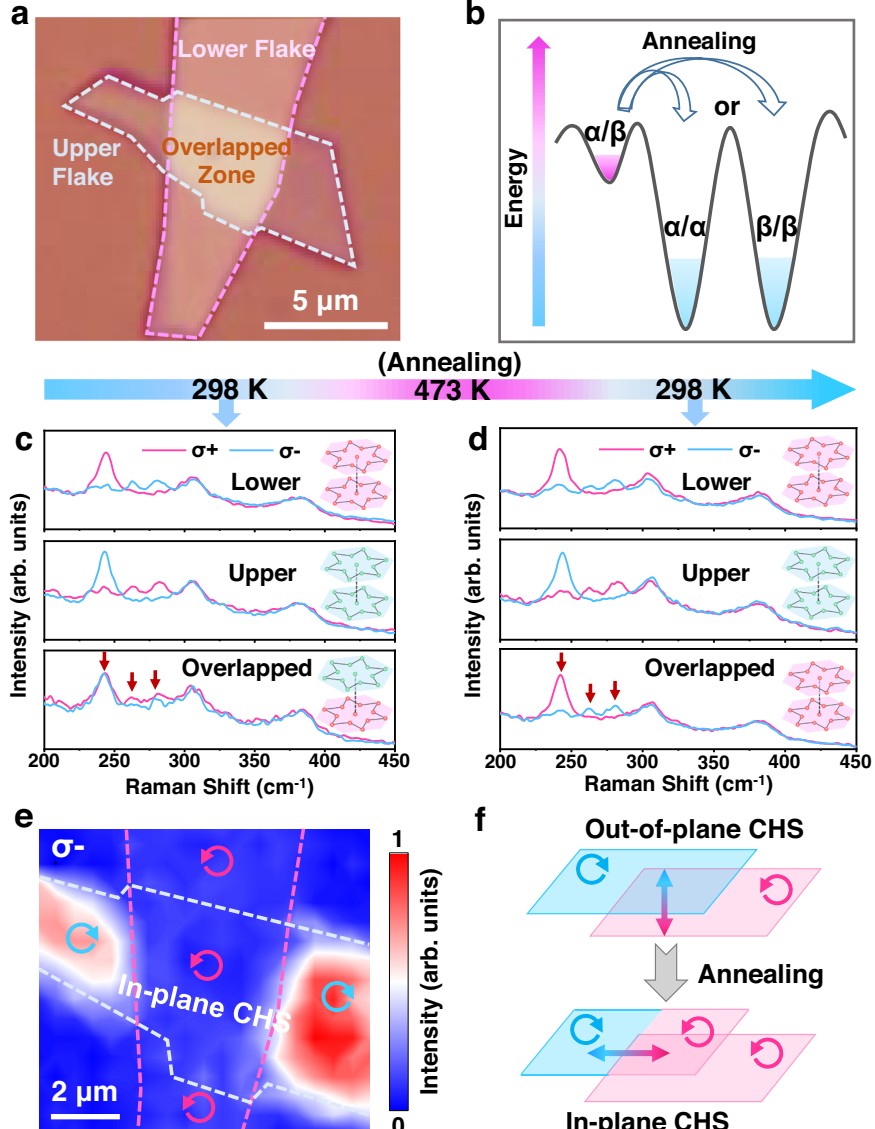

**Fig. 5 | Chirality manipulation of the α/β 1T-TaS₂ homostructure. a** Optical image of the α/β 1T-TaS₂ homostructure. **b** Schematic illustration for the transformation of the chiral stacking configurations. **c** Chiral Raman spectra of the lower flake, upper flake, and overlapped zone of the initial sample. The insets show the schematic diagrams of the corresponding stacking configurations. Pink and green stars represent oppsite chirality. **d** Chiral Raman spectra of the sample after annealing (473 K, 2 hours). The Raman peaks indicated by the red arrows are the $E_g^5$, $E_g^6$, and $E_g^7$ modes, which show apparent chiral Raman response. The wavelength of the excitation laser is 532 nm. **e** Raman intensity mapping image of the $E_g^5$ mode under σ− excitation after the annealing process. The blue and pink arrows indicate the opposite chirality of several parts in the sample, and the in-plane chiral homostructure (CHS) is generated in the upper flake. **f** Schematic diagram of the concept to induce in-plane CHS through the out-of-plane chirality modulation.

lattice deformation generated at the edges of the overlapped zone. Further experiments on other samples also confirm this scenario (Supplementary Fig. 20). Thus chiral domain boundaries and in-plane CHS can be generated in the 2D chiral CDW homostructure, as illustrated in Fig. 5f.

## Discussion

In summary, we revealed the stacking configuration of chiral CDW in 1T-TaS₂ by chiral Raman spectroscopy and established an approach to fabricate in-plane CHS of CDW using the interlayer chirality-locking effect. The reversible chirality switching between two enantiomers and the interlayer chirality-locking effect provides an alternative way to manipulate the chiral CDW, which provides insight into the interlayer interaction of CDW systems. The designed in-plane CHS and the generated chiral domain walls

provide platforms to explore more physical phenomena, and may inspire the future design of devices based on chiral CDW materials. Moreover, some other methods to manipulate chiral phase transition, such as laser pulse and external electric field, may deserve further exploration.

## Methods

### Raman spectroscopy measurement

Raman spectra were measured by a confocal Raman spectroscope (JY Horiba HR800). The excitation laser is 532 nm (2.33 eV), and was focused on the sample through a 50× objective. The excitation power was kept below 1 mW. The circularly polarized light was generated by a polarizer and a quarter-wave plate. Helicity-resolved Raman spectra were measured by rotating a polarizer in the collection light path (Supplementary Fig. 2). The variation of temperature was realized by a

cryogenic chamber (Linkam THMS600), which was refrigerated by liquid nitrogen. After being exfoliated onto the 300 nm $SiO_2$/Si substrate, the 1T-$TaS_2$ sample was subsequently (<30 min) put into the cryostat, and the sample was protected by the argon (Ar) atmosphere to avoid oxidation.

## Scanning tunneling microscopy measurement
The STM images were acquired by the electrochemically etched tungsten tip with Scienta Omicron LT STM system. The measurement was performed by the constant current mode at 77 K. Single crystal surface of bulk 1T-$TaS_2$ was obtained via in situ cleavage at room temperature in preparation chamber (base pressure $1 \times 10^{-9}$ mbar), and the STM experiments were performed in low-temperature chamber (base pressure $2.8 \times 10^{-10}$ mbar). The scanning parameters are: bias voltage $V_b = -0.1$ V, tunneling current $I_t = 5.8$ nA (Fig. 2b); bias voltage $V_b = -0.8$ V, tunneling current $I_t = 2.6$ nA (Fig. 2c).

## SAED measurement
The samples used for SAED in Fig. 2 were prepared by transferring the 1T-$TaS_2$ flakes on 300 nm $SiO_2$/Si substrate onto the holey silicon nitride ($SiN_x$) grids with the assistance of Polypropylene Carbonate (PPC). 1T-$TaS_2$ flakes with opposite chiral Raman responses were selected from those exfoliated onto the 300 nm $SiO_2$/Si substrate, then PPC was spin-coated on the 1T-$TaS_2$/$SiO_2$/Si substrate surface. The PPC/1T-$TaS_2$ films were torn off and then placed onto the $SiN_x$ grids. Then the PPC films were removed by immersing the grids in acetone for 12 hours. SAED was measured at room temperature using aberration-corrected Titan Cubed Themis G2 300 with an accelerating voltage of 300 kV.

## Second harmonic generation measurement
The samples for SHG measurements were prepared by exfoliating the 1T-$TaS_2$ flakes onto the fused silica substrate. Two flakes with similar thickness (~8 nm) showing opposite chiral Raman responses were selected to measure the SHG signal under circularly polarized excitation. The excitation laser with femtosecond pulses (~100 fs, 80 MHz) was generated by a Ti:sapphire oscillator (Spectra-Physics Mai Tai laser), and the SHG signal was collected using the back-scattering configuration. The circularly polarized excitation was realized by placing a quarter-wave plate in the incident light path.

## DFT calculations
The first-principles calculations were implemented in the Vienna Ab Initio Simulation Package[50] (VASP). The pseudopotentials were constructed by the projector augmented wave method[51,52], and the Perdew–Burke–Ernzerhof type[53] within the generalized gradient approximation framework was applied for the exchange-correlation functions. A dimmer of David star with the same or the opposite chirality was constructed in a unit cell. In the case of structural optimization calculations, both the lattice vector and atomic positions were fully relaxed with an energy convergence criterion of $10^{-6}$ eV. A $4 \times 4 \times 4$ k-point mesh and a 520 eV plane-wave energy cutoff were adopted. The DFT-D3 method of Grimme[54] was used to take the van der Waals interaction between layers into account. In the partial radial distribution function calculations, adjacent sulfur atoms in different layers were classified as different atom types to characterize the interlayer structure.

## Data availability
All data supporting the findings of this study are available within the paper and the Supplementary information files. Additional data are available from the corresponding authors upon request. Source data are provided with this paper.

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

## Acknowledgements

The authors thank HORIBA Scientific and Dr. Peng Miao for assistance in Raman spectra measurements. This work was financially supported by the Ministry of Science and Technology of China (Grant Nos. 2018YFA0703502 (L.T.) and 2021YFA1400201 (K.L.)), the National Natural Science Foundation of China (Grant Nos. 51720105003 (J.Z.), 21790052 (J.Z.), 52021006 (J.Z. and L.T.), 21974004 (L.T.), 21972032 (X.Q.), 12025407 (S.M.) and 11934004 (S.M.)), the Strategic Priority Research Program of CAS (Grant Nos. XDB36030100 (J.Z.), XDB330301 (S.M.) and XDB33000000 (K.L.)), the Scientific Instrument Developing Project of the Chinese Academy of Sciences (GJJSTD20200005 (X.Q.)), Youth Innovation Promotion Association of CAS (2022038 (M.L.)), CAS Project for Young Scientists in Basic Research (Grant Nos. YSBR054 (M.L.) and YSBR047 (S.M.)) and the Beijing National Laboratory for Molecular Sciences (BNLMS-CXTD-202001 (J.Z.)).

## Author contributions

Y.Z. and Z.N. conceived the idea. L.T., S.M., and J.Z. supervised the whole work. Y.Z. and S.H. performed the measurements and analysis of Raman spectra. H.H. and K.L. performed the SHG measurement. X.Q., M.L., and X.Q. performed the STM measurement. Y.Y. measured the SAED. Z.N. and S.M. performed the DFT calculations. All authors discussed the results and commented on the manuscript.

## Competing interests

The authors declare no competing interests.
