## [Peer Review File · Nature Communications]

REVIEWER COMMENTS

Reviewer #1 (Remarks to the Author):

The manuscript presents very interesting Raman experiments and theoretical studies of chiral behavior of charge density waves in 1T-TaS₂. In my opinion, the most important and new results are related to demonstration of reversible chirality switching between two enantiomers existing in 1T-TaS₂ in NCCDW. The obtained experimental results are supported by density functional theory (DFT) calculations, which show strong preference of the same-chirality order in subsequent layers. I agree that the presented Raman experiments, together with theoretical calculations show a way to manipulate the chiral CDW in 1T-TaS₂ and provide valuable insight into the interlayer interaction of CDW systems.

In my opinion the presented results are worth for publication in Nature Communications, however the manuscript needs to be revised.

First of all it seems natural to refer to the recently published papers on Raman studies showing the optical activity and chirality of 1T-TaS₂:

- 1) Raman Optical Activity of 1T-TaS₂, E. M. Lacinska et al Nano. Lett. 22, 2385 (2022).
- 2) Visualization of Chiral Electronic Structure and Anomalous Optical Response in a Material with Chiral Charge Density Waves H. F. Yang et. al Phys. Rev. Lett. 129, 156401 (2022).

In particular in the paper by E. M. Lacinska et al Nano. Lett. 22, 2385 (2022) it is presented that measuring Raman response for different angles between linear polarized excitation and scattered light one can obtain additional information about particular modes in NCCDW and CCDW phase, which are not properly described using Raman tensor of C_{3i} group symmetry. The result is quite different for the situation in which the sample is turned, from the situation in which polarization angle of light is changed. The observed polarization rotations are well reproduced with an antisymmetric Raman tensor. It should be worth if the Authors comment on this, as well results presented by H. F. Yang et. al Phys. Rev. Lett. 129, 156401 (2022).

As concerning minor corrections, there is a misprint in Raman intensities under VV and VH configuration, formulae (5) and (6) – $\cos(2\theta)$ should be $\cos^2(\theta)$, and – $\sin(2\theta)$ should be $\sin^2(\theta)$.

Reviewer #2 (Remarks to the Author):

In this paper, authors reports the detection of the chiral CDW phase in thin layer 1T-TaS₂ by Raman spectroscopy, well supported also by STM, SAED and SHG. The 2D mapping of Raman spectroscopy proposed in this paper is quite simple and useful for in-situ detecting the chiral polarity of the CDW in 1T-TaS₂ with spatial resolution. Especially, the demonstration of the cooperative chiral switching in the stacked TaS₂ flakes would give a great insight in understanding the interlayer interaction upon the CDW order in 1T-TaS₂, which is not well clarified yet. Thus, this paper may warrant consideration for publication in Nature Communication. However, the following points need to be revised by the authors before publication.

1) The CDW phase transitions in 1T-TaS₂ exhibit significant hysteresis behavior which depends even on the number of layers of the sample thin film. In Fig. 1, it is claimed that the observed chiral Raman response is specific in the NCCDW and CCDW phases. Authors should put the observed results of the hysteresis behaviors in the transitions, which would justify the above claim too. (Taking account the experimental results shown in Fig.3, authors could put this data in Fig.1) If available, the layer number (thickness) dependence data of the hysteresis behavior (temperature width etc.) is better in understanding the inter-layer interactions in CDW phases in 1T-TaS₂.

2) The thicknesses of the sample thin film in STM, SAED, and SHG is almost similar to Raman spectroscopy, where the cross-sectional profile of the AFM image indicates ca. 15 nm or even the same sample flake was used in all of measurements? Anyway, the

details about the sample thickness should be given also for STM, SAED and SHG. It is also better for readers if the correspondance number of layers are given for each sample thickness.

3) Some of important experimental details are lack in this paper such as a typical time from the exfoliation of the sample to the purge by Ar gas in the cryostat or evacuation in the chamber, base pressure of the STM chamber etc..

4) The difference of the energy of the inter-layer interactions between a/a and a/b stack obtained by the DFT simulation (0.5 eV / star) seems too large for the slight "difference" between 2 types of stacking. It would be almost similar scale to that of the inter-layer interaction of TaS₂ layers itself or even larger. This might come from the limitation of the methods in DFT calculations and any scaling treatments of the calculated results are necessary for comparison with the experimental data. Anyway, authors need to put a justification of the value of "0.5 eV /star" as the difference between a/a and a/b stacking or mention the limitations of the DFT simulation performed in this study.

Reviewer #3 (Remarks to the Author):

Review of paper by Yan Zhao et al.

The paper presents interesting Raman data analysis on the detection of chirality in the stacked charge-ordered phases of 1T-TaS₂. Beyond the use of the Raman method, it demonstrates the usefulness of the technique with measurements on stacked layered structures, which reveal the material's chirality re-stacking properties.

The Raman method is demonstrated to be very powerful here, and will undoubtedly be of interest to the community. While the Raman method itself and application to chirality detection dates to well before Loudon's review paper (o from before 1964, ref. 23), its application to the present material is of interest because the chiral properties of 1T-TaS₂ are currently a hot topic in the field. The method itself however, it is not a breakthrough in any aspect.

What makes the paper interesting, is the use of Raman chirality measurements to detect changes in chirality of stacked multilayered heterostructures upon annealing at high temperature. The idea behind the stacking experiments is clear, and the work is understandably presented. The results are not unexpected, and are in line with current understanding of the material.

In summary, the paper presents the use of an established method that has been applied to such materials before. The interesting part of the work comes from its application to a material that is currently of interest, addressing the problem of chirality switching with temperature. Regarding the suitability for the journal, I'm somewhat reserved. The result is not a breakthrough, yet the results are quite interesting and provide useful information. If there were more detailed experiments on chirality switching and locking, for example by laser heating, where the effect of the strain fields in the central region of the flakes would be revealed, I would be much more enthusiastic.

Minor and technical points:

The calculations do not take into account discommensurations that are present at room temperature. Considering the difference in energies presented in Fig. 4 is only 10%, so I am not convinced that it is more than an indication that there is a possible difference. I would like to see some error bars in the calculation (depending on initial assumptions, the effect of discommensurations, etc), that would make it more convincing.

- Figures 5 c,d could be clearer, particularly the insets.

- The $g(r)$ are not defined.
- The statement regarding 'messy distribution of interlayer sulfur atoms' (line 226) is not acceptable without clarification. Are the authors referring to their sample? If so, I would expect supporting data. If it is a general statement, then I would expect a reference.
- There are some sentences which are difficult to understand. For example the sentence in lines 65-67.
- The last section on in-plane chiral homostructures and the discussion could also benefit from some more concise language.
- Some notations are not defined in the main text, such as A-E' stacking for example, only in the Figure.

The paper would benefit from some improvements of the presentation.

Response to reviewer 1's comments

(All changes made are highlighted in red in the revised manuscript)

First of all, we thank the reviewer for the comments and advice which will help us to improve the quality of our present manuscript. We have addressed these comments point-by-point and carefully revised the manuscript as follows.

“The manuscript presents very interesting Raman experiments and theoretical studies of chiral behavior of charge density waves in 1T-TaS₂. In my opinion. The most important and new results are related to demonstration of reversible chirality switching between two enantiomers existing in 1T-TaS₂ in NCCDW. The obtained experimental results are supported by density functional theory (DFT) calculations, which show strong preference of the same-chirality order in subsequent layers. I agree that the presented Raman experiments, together with theoretical calculations show a way to manipulate the chiral CDW in 1T-TaS₂ and provide valuable insight into the interlayer interaction of CDW systems. In my opinion the presented results are worth for publication in Nature Communications, however the manuscript needs to be revised.”

Reply:

We thank the reviewer for the positive opinion and the insightful comments.

(1) *“First of all it should it seems natural to refer to the recently published papers on Raman studies showing the optical activity and chirality of 1T-TaS₂:*

1) Raman Optical Activity of 1T-TaS₂, E. M. Lacinska et al Nano. Lett. 22, 2385 (2022).

2) Visualization of Chiral Electronic Structure and Anomalous Optical Response in a Material with Chiral Charge Density Waves H. F. Yang et. al Phys. Rev. Lett. 129, 156401 (2022).”

Reply:

We thank the reviewer for kindly recommending us these two papers recently published. We have noticed that both papers show important experimental results about the chirality of charge density waves in 1T-TaS₂. The paper by E. M. Lacinska *et al.* shows the abnormal angle dependence for linearly polarized Raman spectra of 1T-TaS₂, which was attributed to the existence of chiral CDW and the resulted Raman optical activity. The paper by H. F. Yang *et al.* shows the chiral electronic structure and photon-helicity-dependent Raman signal originating from the chiral CDW order. They claimed the chiral signatures and related optical/electronic properties in 1T-TaS₂ for different aspects, which are useful for fully understanding the nature of chiral CDW. According to the reviewer's suggestion, we have added these two references in the main text (*reference 28 and 29, page 4*).

“In particular in the paper by E. M. Lacinska et al Nano. Lett. 22, 2385 (2022) it is presented that measuring Raman response for different angles between linear polarized excitation and scattered light one can obtain additional information about particular modes in NCCDW and CCDW phase, which are not properly described using Raman tensor of C_{3i} group symmetry. The result is quite different for the situation in which the sample is turned, from the situation in which polarization angle of light is changed. The observed polarization rotations are well reproduced with an antisymmetric Raman tensor. It should be worth if the Authors comment on this, as well results presented by H. F. Yang et. al Phys. Rev. Lett. 129, 156401 (2022).”

Reply:

We thank the reviewer for this suggestion. Both papers have observed specific Raman response related to the chiral CDW in 1T-TaS₂, which could be understood by Raman tensor analysis. In the paper by E. M. Lacinska *et al.*, the reported experimental phenomena in linearly and circularly polarized Raman spectra are related to the chiral nature of 1T-TaS₂ in the CCDW phase. The formation of chiral CDW lowers the symmetry of superlattice with the mirror symmetry broken down. The antisymmetric

Raman tensor the authors introduced to explain the anomalous polarized Raman response is a result of the lowered symmetry in the chiral lattice, because the Raman tensor is a deduction from the symmetry analysis on crystal and phonon (*Loudon, R. Adv. Phys. 2001, 50, 813-864*). Also, there may appear non-negligible complex Raman tensor elements for low-symmetry crystals, just as the paper by H. F. Yang *et al.* has mentioned, which may account for the contrast in the circular contrarotating configurations.

Thus we think that the Raman tensor analysis in these two papers are theoretically unified, both based on the classical symmetry analysis. Also, the experimental phenomena in these two papers are consistent with our results, all pointing to the chiral CDW in 1T-TaS₂. The Raman tensor analysis displayed in these two papers would be helpful for us to better understand the chiral Raman response in our present manuscript. We have added relevant discussion in the main text of the revised manuscript (*labeled red in the revised manuscript, page 4*):

“...The out-of-plane A_{1g} Raman modes for the $\sigma^+\sigma^+$ and $\sigma^-\sigma^-$ configurations are almost the same, whereas the intensities of in-plane E_g modes differ remarkably for the $\sigma^+\sigma^-$ and $\sigma^-\sigma^+$ configurations, indicating that the chiral Raman response may correspond to an in-plane chiral structure. **Similar experimental results of chiral Raman response in 1T-TaS₂ have been reported recently, which were well explained by Raman tensor analysis, either by introducing an antisymmetric Raman tensor or including complex Raman tensor elements^{28, 29}.**”

(2) “As concerning minor corrections, there is a misprint in Raman intensities under VV and VH configuration, formulae (5) and (6) – $\cos(2*\Theta)$ should be $\cos^2(\theta)$, and $-\sin(2*\Theta)$ should be $\sin^2(\theta)$.”

Reply:

We thank the reviewer for pointing out this mistake. We are sorry for the misprint, and we have corrected it in the revised manuscript (*Supplementary Information, page 4*),

which is labeled **red** as shown below:

“And the Raman intensities under the VV and VH configurations are:

$$\begin{aligned} I_{A_g}^{VV} &= \left| (\sin \theta \quad \cos \theta \quad 0) \cdot \begin{pmatrix} a & d & e \\ d & b & f \\ e & f & c \end{pmatrix} \cdot \begin{pmatrix} \sin \theta \\ \cos \theta \\ 0 \end{pmatrix} \right|^2 \\ &= (a \sin^2 \theta + 2d \sin \theta \cos \theta + b \cos^2 \theta)^2 \end{aligned} \quad (S5)$$

$$\begin{aligned} I_{A_g}^{VH} &= \left| (\cos \theta \quad -\sin \theta \quad 0) \cdot \begin{pmatrix} a & d & e \\ d & b & f \\ e & f & c \end{pmatrix} \cdot \begin{pmatrix} \sin \theta \\ \cos \theta \\ 0 \end{pmatrix} \right|^2 \\ &= [d(\cos^2 \theta - \sin^2 \theta) + (a - b) \sin \theta \cos \theta]^2 \end{aligned} \quad (S6)”$$

Response to reviewer 2's comments

(All changes made are highlighted in red in the revised manuscript)

“In this paper, authors reports the detection of the chiral CDW phase in thin layer 1T-TaS₂ by Raman spectroscopy, well supported also by STM, SAED and SHG. The 2D mapping of Raman spectroscopy proposed in this paper is quite simple and useful for in-situ detecting the chiral polarity of the CDW in 1T-TaS₂ with spatial resolution. Especially, the demonstration of the cooperative chiral switching in the stacked TaS₂ flakes would give a great insight in understanding the interlayer interaction upon the CDW order in 1T-TaS₂, which is not well clarified yet. Thus, this paper may warrant consideration for publication in Nature Communications. However, the following points need to be revised by the authors before publication.”

Reply:

We thank the reviewer for positive opinion and comments on this manuscript.

(1) “The CDW phase transitions in 1T-TaS₂ exhibit significant hysteresis behavior which depends even on the number of layers of the sample thin film. In Fig. 1, it is claimed that the observed chiral Raman response is specific in the NCCDW and CCDW phases. Authors should put the observed results of the hysteresis behaviors in the transitions, which would justify the above claim too. (Taking account the experimental results shown in Fig.3, authors could put this data in Fig.1) If available, the layer number (thickness) dependence data of the hysteresis behavior (temperature width etc.) is better in understanding the inter-layer interactions in CDW phases in 1T-TaS₂.”

Reply:

We thank the reviewer for this helpful suggestion. We agree that the hysteresis behavior of CDW phase transition would be important and should be included in this manuscript.

Therefore, we performed new measurements about the temperature dependence of chiral Raman response. We plotted the variation of circular polarization degree of E_g^6 mode with temperature in the heating and cooling processes, which shows clear hysteresis behavior in the NCCDW-ICCDW transition region. As shown in the Figure R1 below, for CCDW and NCCDW phases, the E_g^6 mode shows clear chiral Raman response, which only appears in σ^- polarized excitation with the polarization degree ($\rho = (I_{\sigma^-} - I_{\sigma^+}) / (I_{\sigma^-} + I_{\sigma^+})$) to be 1. Whereas for the ICCDW phase, the chiral Raman response disappears with $\rho = 0$. With the assistance of the above data, we demonstrate the correlation between chiral Raman response and chiral CDW phase, that is, the chiral Raman response only appears in NCCDW and CCDW phase. Taking account of the whole structure of the manuscript, we have put this data in *Supplementary Information Fig. S11*, as shown in the following.

Fig. R1 (Fig. S11) Hysteresis behavior of chiral Raman response. Variation of circular polarization degree ($\rho = (I_{\sigma^-} - I_{\sigma^+}) / (I_{\sigma^-} + I_{\sigma^+})$) of E_g^6 mode for 1T-TaS₂ flakes with the thickness of **a**, 13 nm; **b**, 20 nm; **c**, 115 nm in the heating (red) and cooling (blue) processes. For NCCDW phase, the E_g^6 mode shows clear chiral Raman response, which only appears in σ^- polarized excitation with $\rho = 1$. Whereas for the ICCDW phase, the chiral Raman response disappears with $\rho = 0$. Thus the hysteresis behavior of chiral Raman response is consistent with the NCCDW-ICCDW transition process, and the thickness dependence for the hysteresis behavior in the NCCDW-ICCDW transition is somewhat robust, which is consistent with the previous report¹¹.

(2) *“The thicknesses of the sample thin film in STM, SAED, and SHG is almost similar to Raman spectroscopy, where the cross-sectional profile of the AFM image*

indicates ca. 15 nm? or even the same sample flake was used in all of measurements? Anyway, the details about the sample thickness should be given also for STM, SAED and SHG. It is also better for readers if the correspondance number of layers are given for each sample thickness.”

Reply:

We thank the reviewer for this helpful suggestion. We did not use the same sample for all these measurements, because we need to prepare specific samples fit for the requirements of each characterization method. Thus the sample thickness may vary slightly for these measurements. For SAED measurement, the sample was firstly exfoliated on SiO₂/Si substrate and then transferred to the holey silicon nitride (SiN_x) grids. The sample is shown below (extracted from Figure S11), and the thickness is approximately 15 nm.

Figure R2. 1T-TaS₂ sample for SAED measurement.

For SHG measurement, the 1T-TaS₂ sample was exfoliated on the fused silica substrate as shown below, and the thickness is approximately 8 nm.

Figure R3. 1T-TaS₂ sample on fused silica substrate for SHG measurement

For STM measurement, we used the bulk samples because we could not find the few-

layered samples due to the low optical contrast using the objective with long working distance equipped to the STM chamber. It is also more convenient for cleavage to find a clean surface. The purpose here is to show the two chiral CDW states existing in 1T-TaS₂ through the STM results, which has also been claimed by C. H. Wen *et al* (*Phys. Rev. Lett.* 2021, 126, 256402) and H. F. Yang *et al* (*Phys. Rev. Lett.* 2022, 129, 156401) previously. The thickness variation would not affect this conclusion.

As for the layer number, we noticed that the thicknesses of 0.6 nm or 0.65 nm for a single layer have been reported in previous papers (*Nat. Nanotechnol.* 2015, 10, 270-276; *J. Phys. Chem. C* 2020, 124, 27176–27184). Thus we evaluate that the sample we used with a thickness of 15 nm may correspond to 23~25 layers. However, the thickness we measured from atomic force microscope (AFM) may vary from the literature due to the substrate roughness and the vertical resolution of AFM etc. Thus we did not give accurate numbers of layers in the present manuscript, but use the thickness to define the samples, just as in several previous papers (*Sci. Rep.* 2014, 4, 7302; *Chem. Mater.* 2016, 28, 7613–7618).

According to the reviewer's suggestion, we have added the thickness information of these sample for STM, SAED and SHG measurements in the main text (labeled red in the revised manuscript), as shown below:

“**Fig. S11 a.** Optical picture of the mechanically exfoliated 1T-TaS₂ flake **with a thickness of approximately 15 nm** on 300 nm SiO₂/Si substrate.” (*Revised Supplementary Information, page 11*)

“The samples for SHG measurements were prepared by exfoliating the 1T-TaS₂ flakes onto the fused silica substrate. **Two flakes with similar thickness (~ 8 nm) showing opposite chiral Raman responses** were selected to measure the SHG signal under circularly polarized excitation.” (*Revised main text, page 15*)

“In Fig. 2b and 2c, we show the STM images obtained at 77 K for the two enantiomers, **which is measured in bulk 1T-TaS₂**, revealing the chiral CDW states corresponding to

the chiral superlattices.” (*Revised main text, page 6*)

(3) “*Some of important experimental details are lack in this paper such as a typical time from the exfoliation of the sample to the purge by Ar gas in the cryostat or evacuation in the chamber, base pressure of the STM chamber etc.*”

Reply:

We thank the reviewer for this comment. We have added these detailed information in the corresponding Methods parts in main text (labeled **red** in the revised manuscript), as shown below:

“The variation of temperature was realized by a cryogenic chamber (Linkam THMS600) which was refrigerated by liquid nitrogen. **After being exfoliated onto the 300 nm SiO₂/Si substrate, the 1T-TaS₂ sample was subsequently (less than 30 minutes) put into the cryostat, and the sample was protected by the argon (Ar) atmosphere to avoid oxidation.**” (*Main text, page 14*)

“The measurement was performed by the constant current mode at 77 K. **Single crystal surface of bulk 1T-TaS₂ was obtained via in situ cleavage at room temperature in preparation chamber (base pressure 1×10^{-9} mbar), and our STM experiments were performed in low temperature chamber (base pressure 2.8×10^{-10} mbar).**” (*Main text, page 15*)

(4) “*The difference of the energy of the inter-layer interactions between a/a and a/b stack obtained by the DFT simulation (0.5 eV/star) seems too large for the slight “difference” between 2 types of stacking. It would be almost similar scale to that of the inter-layer interaction of TaS₂ layers itself or even larger. This might come from the limitation of the methods in DFT calculations and any scaling treatments of the calculated results are necessary for comparison with the experimental data. Anyway,*

authors need to put a justification of the value of “0.5 eV /star” as the difference between a/a and a/b stacking or mention the limitations of the DFT simulation performed in this study.”

Reply:

We thank the reviewer for pointing out this question. In Fig. 4c, the interlayer interaction for the same-chirality stacking is about 4.4 eV/star. The energy difference of 0.5 eV/star only accounts for 11% of the binding energy. It has been reported that the typical interlayer coupling strength of graphite is about ~100 meV/atom (for example, *Chen, X., Tian, F., Persson, C. et al. Interlayer interactions in graphites. Sci Rep 3, 3046 (2013)*). Our DFT result (4.4 eV/star \approx 113 meV/atom) is very close to this range, and the energy difference of 0.5 eV/star corresponds to 13 meV/atom, which we think is reasonable. According to the reviewer’s suggestion, we have put a justification of the value of “0.5 eV /star” as the difference between α/α and α/β stacking in the revised manuscript, as shown below:

“We listed the calculated interlayer binding energies (E_{ib}) and total energies (E_{total}) of these possible stacking orders in Table S1, with $E_{ib} = (\sum E_{layer} - E_{total})/N_{cell}$. Usually, the typical interlayer binding energy of graphite is about ~100 meV/atom³⁰. Our DFT results of binding energy (4.4 eV/star \approx 113 meV/atom) is very close to this number, indicating it is a reasonable value.” (*Supplementary Information, page 16*)

“We list the calculated binding energies in Table S2, among which the A-E’ stacking shows the highest binding energy and is the most stable one among all the listed stacking configurations. It’s worth noting that the energy difference of about 0.5 eV/star accounts for ~11% of the interlayer binding energy of α/α stacking.” (*Supplementary Information, page 18*)

Response to reviewer 3's comments

(All changes made are highlighted in red in the revised manuscript)

“The paper presents interesting Raman data analysis on the detection of chirality in the stacked charge-ordered phases of 1T-TaS₂. Beyond the use of the Raman method, it demonstrates the usefulness of the technique with measurements on stacked layered structures, which reveal the material's chirality re-stacking properties.

The Raman method is demonstrated to be very powerful here, and will undoubtedly be of interest to the community. While the Raman method itself and application to chirality detection dates to well before Loudon's review paper (o from before 1964, ref. 23), its application to the present material is of interest because the chiral properties of 1T-TaS₂ are currently a hot topic in the field. The method itself however, it is not a breakthrough in any aspect.

What makes the paper interesting, is the use of Raman chirality measurements to detect changes in chirality of stacked multilayered heterostructures upon annealing at high temperature. The idea behind the stacking experiments is clear, and the work is understandably presented. The results are not unexpected, and are in line with current understanding of the material.

In summary, the paper presents the use of an established method that has been applied to such materials before. The interesting part of the work comes from its application to a material that is currently of interest, addressing the problem of chirality switching with temperature. Regarding the suitability for the journal, I'm somewhat reserved. The result is not a breakthrough, yet the results are quite interesting and provide useful information. If there were more detailed experiments on chirality switching and locking, for example by laser heating, where the effect of the strain fields in the central region of the flakes would be revealed, I would be much more enthusiastic.”

Reply:

We thank the reviewer for the insightful comment. We agree that chirality switching and locking are interesting topics which deserve more exploration, especially if we can apply more external stimuli such as laser heating and electric field to manipulate the chiral CDW. We noticed that several groups have reported the pulsed laser induced phase transition (*Sci. Adv.* 2018, 4, eaau5501; *Sci. Adv.* 2015, 1, e1500168.), thus using the pulsed laser heating to control the chiral phase switching would be a promising method. We have not explored this topic due to the limitation of our experimental setups at the present stage; nevertheless, we believe that these topics deserve further investigation in the future. And also, we hope our work will inspire more research in this field. We have added the following outlook in the Discussion section of our revised manuscript, as shown below.

“Moreover, some other methods to manipulate chiral phase transition, such as laser pulse and external electric field, may deserve further exploration.” (*Revised main text, page 14*)

(1) *“The calculations do not take into account discommensurations that are present at room temperature. Considering the difference in energies presented in Fig. 4 is only 10%, so I am not convinced that it is more than an indication that there is a possible difference. I would like to see some error bars in the calculation (depending on initial assumptions, the effect of discommensurations, etc), that would make it more convincing.”*

Reply:

We thank the reviewer for the insightful suggestion. Due to unsurmountable computational costs of current DFT calculations for aperiodic extended systems, we could not perform calculations directly for nearly commensurate (NC) and incommensurate (IC) phases. However, commensurate (C) calculations could in principles capture the stacking features in NC phase. First of all, the energy difference, 0.5 eV/star (i.e., 13 meV/atom), is much larger than the precision of typical DFT

calculations today (~ 1 meV/atom). This value also reaches half of the thermal energy at the room temperature (25 meV/atom). Thus we believe that the energy difference is reasonable and is sufficient to explain our experimental results.

(2) “*Figures 5 c, d could be clearer, particularly the insets*”

Reply:

We thank the reviewer for this suggestion. We have improved the resolution of insets of Figure 5 c, d, and have provided high-resolution figures, as shown below:

Figure 5. Chirality manipulation in the α/β 1T-TaS₂ homostructure.

(3) “*The g(r) are not defined.*”

Reply:

We thank the reviewer for pointing it out. In statistical mechanics, the radial distribution function $g(r)$ describes how density varies as a function of distance from a reference particle. Consider a system of N atoms with density ρ , then $\rho g(r)dr$ is the probability of observing a second atom in dr given that there is an atom at the origin of r . It gives

$$\int_0^{\infty} \rho g(r) 4\pi r^2 dr = N - 1$$

The function $g(r)$ can also be thought of as the factor that multiplies the bulk density ρ to give a local density $\rho(r) = \rho g(r)$ about some fixed particles. It is also called a correlation function, since if the particles were independent of each other, $\rho(r)$ would simply equal ρ , and so the factor $g(r) \approx 1$. Usually in a solid the atoms are arranged in a regular repeating order, and this lead to sharp peaks in the $g(r)$ curves. (More definition and deduction see McQuarrie, Statistical Mechanics, chapter 13-2 (1976).)

According to the reviewer's suggestion, we added the definition of $g(r)$ mentioned above in the revised manuscript, as labeled in red in the *revised Supplementary Information, page 19*. We also give a brief definition in the *revised main text, page 10*, as shown below:

“Moreover, we use the partial radial distribution functions ($g(r)$), **which describes how particle density varies as a function of distance from a reference particle**, to reflect the distribution of sulfur atoms between layers and to further understand the origin of the E_{ib} difference **(see more detail of $g(r)$ in Supplementary Information Section IV).**”

(4) “The statement regarding ‘messy distribution of interlayer sulfur atoms’ (line 226) is not acceptable without clarification. Are the authors referring to their sample? If so, I would expect supporting data. If it is a general statement, then I would expect a reference.”

Reply:

We are sorry for not clarifying this sentence. The “messy distribution of interlayer sulfur atoms” refers to the atomic distributions of sulfur atoms between successive layers for opposite-chirality stacking. As explained in question (3), $g(r)$ reflects the particle density variation as a function of distance from a reference particle. We can see discrete sharp peaks of the same-chirality A-E stacking in Fig. 4d, implying highly-ordered regular atomic distributions. In contrast, in opposite-chirality A-E' stacking, there are gentle curves oscillating around $g(r) = 1$, indicating weaker correlations between sulfur atoms. Sulfur atom should distribute more randomly, which we referred to as “messy”. In order to avoid misunderstanding, we changed the sentence to “... which indicates that sulfur atoms distribute more randomly.” as labeled in red in the *revised Supplementary Information, page 10*.

(5) *“There are some sentences which are difficult to understand. For example the sentence in lines 65-67.”*

Reply:

We thank the reviewer for this comment. We have improved the description in the present manuscript to make the sentences more clear. As for the sentence in lines 65-67, we have revised them as follows (labeled red in the revised manuscript, page 3):

“Utilizing the energy preference for same-chirality stacking, we realized the fabrication of in-plane CHS by vertically stacking two 1T-TaS₂ flakes with opposite chirality. The overlapped zone will transform from opposite-chirality stacking to same-chirality stacking after annealing, whereas the chirality of other parts of the flakes will not be affected, enabling the formation of chiral domain walls in one flake.”

(6) *“The last section on in-plane chiral homostructures and the discussion could also benefit from some more concise language.”*

Reply:

We thank the reviewer for this suggestion. We have checked the sentences in the section of in-plane chiral homostructures to make it more concise. For example,

1. We have simplified the sentence “After annealing, the α/β stacking can transform to either α/α or β/β stacking, that is, the chiral phases of two flakes in the overlapped zone can transform to the same one.” to “**After annealing, the α/β stacking in the overlapped zone can transform to the more stable α/α or β/β stacking.**”
2. We have replaced the sentence “The Raman spectra of the primitive homostructure (Fig. 5c) reveal that the lower and upper flakes show opposite chiral Raman response, and the chiral Raman signal of the overlapped zone vanishes. After annealing at 473 K for two hours, the upper and lower flakes remain their initial chirality whereas the chiral Raman response appears in the overlapped zone (Fig. 5d) and becomes the same as the lower flake, indicating that the chiral phase of the upper flake is transformed specifically into the same as the lower flake in the overlapped zone.” by “**As shown in Fig. 5c and 5d, the chiral Raman response of the overlapped zone vanishes for the primitive homostructure, whereas it appears after annealing at 473 K for two hours, which indicates that the opposite-chirality stacking has transformed to the same-chirality stacking for the overlapped zone.**”
3. We have changed the sentence “Moreover, we note that the chiral phase transition only occurs in the overlapped zone, thus a domain boundary between α and β phases is generated in the upper flake along the edge of the lower flake” to “**Moreover, we note that a domain boundary between α and β phases is generated along the edges of the overlapped zone**”.
4. Also, we have deleted some sentences, such as “Thus we provide an approach to design the 2D chiral CDW homostructure through the interlayer chirality-locking effect, and may inspire more explorations on novel physical properties.”

See the revised full text of the last section on in-plane chiral homostructures and the discussion in the *revised main text*, page 12.

(7) “Some notations are not defined in the main text, such as A-E’ stacking for example, only in the Figure.”

Reply:

We thank the reviewer for this suggestion. We have added the definition of A-E’ stacking in the Main text and Supplementary Information (*labeled in red in the revised manuscript*), as shown below:

“Fig. 4d displays the calculated $g(r)$ for the same-chirality A-E stacking (**the most stable same-chirality stacking configuration with a stacking vector $\mathbf{T}_s=-2\mathbf{a}+\mathbf{c}$, see detailed definition in Supplementary Information Section IV and Fig. S15**) and opposite-chirality A-E’ stacking (**see detailed definition in Supplementary Information Section IV and Fig. S16**).” (*Main text, page 10*)

“For example, A-E’ stacking means that the central Ta site of the star-of-David in one of the two adjacent layers are aligned with $\mathbf{T}_{S1}=\mathbf{c}$, and the Ta in the other layer is stacked with a relative displacement $\mathbf{T}_{S2}=-2\mathbf{a}+\mathbf{c}$. The notation X' is used to differentiate from X ($X=A, B, C, D, E$) used for the same-chirality stacking.” (*Supplementary Information, page 17*)

REVIEWER COMMENTS

Reviewer #1 (Remarks to the Author):

As I have already emphasized, the work presents new results that are of great interest to the wide community dealing with 2D materials. The revised version of the manuscript takes into account all my suggestions and recommendations. In my opinion, the article is ready for publication in Nature Communications in its current form.

Reviewer #2 (Remarks to the Author):

Based on reviewers' comments, authors have revised their manuscript well, especially in the part related to the hysteresis behavior of the CDW transitions in TaS₂ which authors have performed new measurements, and the justification of their calculation results. Thus, I think this paper could be worth for publication in Nature Communication.

Reviewer #3 (Remarks to the Author):

I have considered the reponse of the authors to my comments and those of the other authors.

Unfortunately, I still don't see a breakthrough of general interest in the revised paper.

I am also not convinced by the arguments regarding the accuracy of the DFT calculations. The claimed accuracy of 1meV/atom is fictitious: it compares only calculations with themselves, not with any experiment. (The total energy is not an experimentally measurable quantity). Considering the presence of domains, stains, impurities, defects and imperfections in the real experiment, the accuracy claim applied to the experiment is unsubstantiated.

Assuming the discussion of the DFT is revised, the might be of interest to Raman spectroscopists and the community working on this material.

Response to reviewer 3's comments

"I have considered the response of the authors to my comments and those of the other authors.

Unfortunately, I still don't see a breakthrough of general interest in the revised paper.

I am also not convinced by the arguments regarding the accuracy of the DFT calculations. The claimed accuracy of 1meV/atom is fictitious: it compares only calculations with themselves, not with any experiment. (The total energy is not an experimentally measurable quantity). Considering the presence of domains, strains, impurities, defects and imperfections in the real experiment, the accuracy claim applied to the experiment is unsubstantiated.

Assuming the discussion of the DFT is revised, the might be of interest to Raman spectroscopists and the community working on this material."

Response:

We thank the reviewer for pointing out the important concern. First of all, we agree that the accuracy of DFT should be improved considering the presence of domains, strains, impurities, etc. However, due to the unaffordable computational cost, we cannot include these factors in our calculations at the present stage. To make this point clear, we have added the discussion about the limitation of DFT in the revised manuscript (labeled in red), as shown below:

"Finally, it is worth pointing out that although our DFT calculations explain the experimental observations well, we cannot directly compare the theoretical results with experiment since the total energy of a given material configuration is not experimentally measurable. Besides, the present DFT calculations adopt an ideal model for the CCDW phase, whereas the presence of domains, strains, impurities, defects and imperfections are not included due to the unsurmountable computational costs. Further comprehensive study is needed to make the DFT results more precise and directly comparable with the experiments." (Main text, page 11)

Also, the reviewer mentioned that the DFT results of total energy cannot be compared with the experiment directly, which makes the DFT accuracy unevaluable. It is true that we cannot measure the total energy experimentally, but we can still find more supporting data from the previous literatures to verify the reliability of our DFT results.

Firstly, Lee *et al.* have calculated the stacking energy of various stacking orders of 1T-TaS₂, with the total energy difference ranging from 1.1 meV/star to 60.7 meV/star (*Phys. Rev. Lett.* **122**, 106404 (2019)). The authors demonstrated that this energy difference is sufficient to justify that the L and AL stacking (corresponding to the E and

AE stacking in our manuscript) is far more stable than the other configurations. The related content is extracted as below:

“The total energy of the system varies up to ~ 60 meV/star depending on the stacking configuration. The equilibrium layer spacing also changes significantly up to $\sim 1\%$, producing large changes in the van der Waals energy of ~ 170 meV/star. The calculated total energies show that two configurations are exceptionally stable, the AL and L stacking configurations. They have a small energy difference of 1.1 meV/star but are far more stable than the other configurations (by > 30 meV/star).”

Besides, similar statement has also been reported by Jiang *et al.* (*Phys. Rev. B* **104**, 075147 (2021)), which points out that the energy difference of 31 meV/f.u. in 1T-TaS₂ can make the CCDW phase more stable than normal phase, as shown below:

“Comparing the stability between CCDW and undistorted 1T phase of monolayer TaS₂, we find that the lattice reconstruction can lower the energy by about 31 meV/f.u. against undistorted phase, indicating that the CCDW phase is more stable than the normal 1T phase.”

Our DFT results of total energy difference for the same-chirality stacking ranging from 3.4 meV/star to 58.6 meV/star (as shown in Table S1 in Supplementary Information) are similar with the results reported by Lee *et al.* Moreover, our calculated energy difference of ~ 0.5 eV/star between opposite-chirality stacking and same-chirality stacking is much larger than the criteria commonly adopted, which makes our explanation on the more stable same-chirality stacking through the energy difference reasonable.

What’s more, the energy difference in chirality stacking has a significant impact on layer spacings. We list the layer spacing of several same-chirality and opposite-chirality stacking configurations in Table I below, the opposite-chirality stacking usually expand 5%-7% in the out-of-plane direction, much larger than that of the same-chirality stacking ($< 2\%$), implying a stronger repulsion between layers in the former, therefore the opposite-chirality stacking is less stable.

	AA	AB	AC	AD	AE	AA'	AB'	AC'	AD'	AE'
Δc (%)	0.95	0.94	0.32	1.12	0.00	6.11	4.95	6.31	5.45	5.57

TABLE I. Equilibrium layer spacing as a function of the CDW stacking configuration, which are compared with the most stable AE stacking

REVIEWERS' COMMENTS

Reviewer #3 (Remarks to the Author):

The authors have responded satisfactorily to my comments regarding the misleading claims based on DFT calculations. I am comfortable with their response written to the referee's comments, and clearly the authors appreciate the problem.

However, the new paragraph in the main text is still misleading and self-contradictory: One cannot say that "DFT calculations explain the experimental observations well" and then continue to say in the same sentence that "one cannot directly compare the theoretical results with experiment". I urge the authors to be more concise in their statement. Only that way can the readers be convinced that the presented treatise is worth taking seriously.